chemical engineering/green chemistry/
computer-aided design

distillation process, biopolyol separation, response surface methodology, economic optimization

**Author for correspondence:**
Tao Chen
e-mail: chentao123@tju.edu.cn

# A comprehensive economic optimization methodology of divided wall columns for biopolyol separation

Tao Chen[1,2], Lingjuan Lv[1,2], Yuanzhi Chen[1,2] and Peng Bai[1,2]

[1]Department of Pharmaceutical Engineering, School of Chemical Engineering and Technology, Tianjin University, Tianjin 300350, People's Republic of China
[2]Key Laboratory of Systems Bioengineering (Ministry of Education), Tianjin University, Tianjin 300072, People's Republic of China

Global energetic and environmental crises have attracted worldwide attention in recent years. Biomass is an important direction of development for limiting greenhouse gas emissions and replacing fossil fuel. As downstream products of biomass, some industrially valuable polyols are costly to separate via conventional distillation due to their near volatility. The use of fully heat-integrated divided wall columns (DWCs), which carry energy and equipment investment savings, is a promising technique for purifying biopolyol products. However, the design of DWCs is complex because of the greater freedom of units, so the optimization of all variables is essential to minimize the cost of separation. A response surface methodology (RSM)-based Box–Behnken design (BBD) was proposed and applied to study the interactions between groups of factors and the effects of variables on total annual cost (TAC) savings. The optimization of global variables with RSM was confirmed to be a powerful and reliable method, and the TAC savings reached 41.09% compared to conventional distillation. In short, efficient design, lower costs and energy savings for polyol separation will promote the wide application of environmentally friendly biopolyol.

## 1. Introduction

With global efforts to reduce emissions and the increasing depletion of fossil fuels, more attention has been paid to producing fuels from biomass. It has been reported that the per cent contribution of biofuels to the total road transport fuel demand was 3% in 2013 and is estimated to grow to 8% by 2035 [1]. Biodiesel is obtained by direct transesterification of vegetable oils or tallows [2]; the by-product glycerol is generated at a rate of approximately 10% during

**Table 1.** Nomenclature.

| | |
|---|---|
| $\beta$ | regression coefficients of the independent variables |
| $\varepsilon$ | error terms |
| D | top product |
| N1 | top section tray number |
| N2 | tray number of prefractionator |
| N3 | tray number of side column |
| N4 | bottom section tray number |
| $q_F$ | thermal feed states |
| R | reflux ratio |
| $r_F$ | ratio of feed tray number to total tray number in prefractionator |
| $r_S$ | ratio of side stream tray number to total tray number in side column |
| S | side stream |
| $S_L$ | liquid split ratio |
| $S_G$ | gas split ratio |
| T1 | first column of conventional distillation |
| T2 | second column of conventional distillation |
| W | bottom product |
| x | levels of the independent variables |
| indexes | |
| C | conventional distillation |
| D | divided wall column |
| i | linear coefficient |
| ii | the squared effect |
| ij | interaction effect |
| j | quadratic coefficient |

the production of biodiesel [3]. The production of crude glycerol from biodiesel has increased dramatically in the past decade worldwide, increasing from 52 million kg in 2016 to 295 million kg in 2018 [4]. This by-product glycerol from biodiesel production has led to a substantial surplus in glycerol supply and caused a significant drop in price for both crude and purified glycerol in the past years [5]. The economically valuable usage of this biomass co-product can be achieved via the chemical route of glycerol hydrogenolysis. The main products are polyols, including ethylene glycol (EG), 1,2-propylene glycol (1,2-PG) and 1,3-propylene glycol (1,3-PG) [6–8], which are important chemical solvents and raw materials widely used in the cosmetic, pharmaceutical, resin and polyester polytrimethylene terephthalate industries, among others [4,9,10]. However, large amounts of energy are consumed in conventional distillation to separate these polyols because of their near volatility. As a result, one promising way to address this challenge is to employ cutting-edge intensification technologies such as divided wall columns (DWCs) (table 1).

DWCs, in which both columns are located in a single-shell, and three-product streams are collected by the insertion of a dividing wall, have been studied since 1949 [11]. DWCs can not only save space and investment requirements because of the reduced number of columns and heat exchangers (figure 1) but also reduce energy costs by reducing the remixing effect [12] of streams; the economic savings can reach 30% [13]. Therefore, the application of DWCs to separate near-volatile biopolyol would save substantial costs relative to that of conventional approaches. However, fully heat-integrated DWCs have more degrees of freedom than conventional columns, and the design of such units is complex. Many researches about the short-cut design of DWCs were based on the Fenske–Underwood–Gilliland–Kirkbride (FUGK) method aimed at minimum vapour flow rate [12,14,15]. Besides, recent articles have reported other systematic methods and models for DWC design: a rigorous model validated in

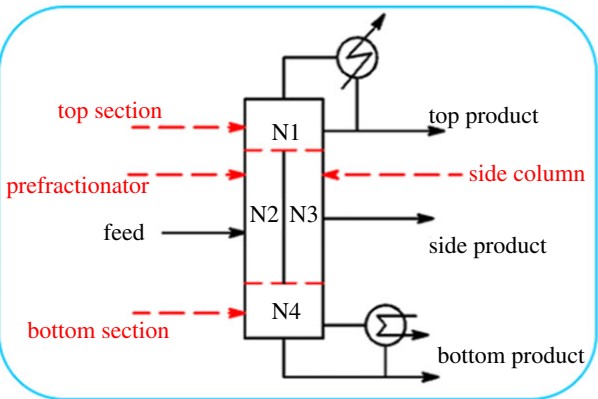

**Figure 1.** Schematic diagram of a DWC.

the experiment was used for the scaled design [16]; coordinate descent methodology with a random research [17] and optimization-based design approach with an automated initialization procedure were employed to determine variables of DWCs during the initial design [18]. As a result, not only the short-cut design but further optimization for these complex chemical units should be attached great importance for separation mission, and different values of these variables could have an unequal effect on the energy or capital costs of the separation process. An effective optimization method is needed to obtain the best combination of variables for DWCs.

Since the optimization of DWCs is a mixed integer nonlinear programming problem (MINLP), which cannot be solved by commercially available process simulators, various methods have been suggested for optimizing DWCs. Dünnebier and Pantelides [19] proposed a local optimization code—CONOPT—which was interfaced with the gPROMS process modelling tool. This method can provide initial design parameters for rigorous analysis and obtain relatively optimal results but cannot achieve the exhaustive optimization of DWCs. Gómez-Castro et al. [20] proposed an optimization technique using genetic algorithms for an alternative design with a post-fractionator instead of a prefractionator, and multi-objective genetic algorithms were also used to optimize heat pump-assisted reactive DWC considering the economic and thermodynamic efficiency performances [21]. To handle disturbance and implementation error of DWCs, Luo et al. [22] developed an online optimization method assisted with steady-state analysis. In addition, a sequential quadratic method using the mixed integer linear programming problem [23] or MINLP [24] approach was also used for conceptual design of DWCs. However, this kind of method is difficult to implement. Premkumar & Rangaiah [25] and Long et al. [26] optimized only one variable at a time while keeping others constant, and this simple method did not consider interactions between variables. For optimization of DWCs, especially for separating near-volatile polyols, a large number of structural and process variables should be taken into account to reach the maximum cost savings. These variables interact with each other and should be optimized simultaneously to obtain the optimal design.

Response surface methodology (RSM) is a useful technique in process optimization studies for building regression models, optimizing a response and identifying the relationships of several variables and their interactions [27]. RSM uses quantitative data from a specified experimental design to solve multivariable problems, and its design is clearer and easier than those of other methods. As a result, RSM is generally used in optimization research on multivariable systems, including the simulation of structural or process parameters in DWCs [28,29]. However, previous studies did not achieve global optimization or consider the interactions between key variables.

This work proposes a simple and efficient method to optimize the variables of a DWC with full consideration of the interactions between key factors. The design of a DWC for biomass polyol separation was studied with the aim of minimizing the total annual cost. Finally, an actual simulation was run to validate the fitness of the regression models.

# 2. Design and optimization methods

## 2.1. Design

The conversion rate of glycerol and the composition of generated three polyol in hydrogenolysis reaction mainly depend on the catalyst and reaction conditions [5,6,8,30]. In this research, the separation study

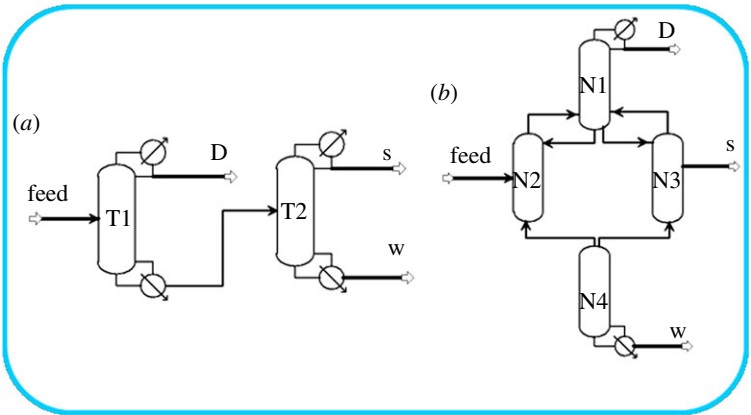

**Figure 2.** Conventional distillation sequence (*a*) and an equivalent DWC of four column for initial design (*b*).

was based on the product from the industrially valuable biomass conversion using Cu/Ni-based catalyst [8,31,32] with high conversion rate of glycerol, and the composition of main biopolyol products from glycerol hydrogenolysis was chosen to be 1,2-PG (approx. 30 mol%), EG (approx. 40 mol%) and 1,3-PG (approx. 30 mol%), because it was challenging but meaningful to obtain three purified products with close quantity through distillation. As a result, these polyol mixtures were separated through DWC and conventional distillation sequences to compare their total annual costs (TACs). Physical–chemical properties for polyol were taken from the Aspen Plus component database. The NRTL [33] model was chosen to describe the phase behaviour of this polyol system and all binary interactive parameters from Aspen Plus were listed in electronic supplementary material, table S1 and consisted with reports 34–36. Moreover, polyol has relatively high boiling point (1,2-PG: 461.3 K; EG: 470.8 K; 1,3-PG: 483.7 K) at 1 atm and its boiling needs expensive high-grade heat source during distillation process. On another hand, as shown in electronic supplementary material, figure S1, when the liquid mole fraction of EG is lower than around 0.2, the relative volatility of 1,2-PG/EG and EG/1,3-PG was lower at vacuum condition than that at high pressure, and this decline becomes relatively heavy from 15 to 5 kPa, which is bad for polyol's separation. As a result, considering both relatively lower heat requirement and higher separation efficiency, 15 kPa was chosen as the operation pressure of distillation unit. Moreover, the vapour–liquid equilibrium data of polyol [34–36] are presented graphically in electronic supplementary material, figure S2 which indicated that the boiling points of 1,2-PG, EG and 1,3-PG were 406.3, 415.2 and 431.7 K, respectively, at operation pressure.

The TAC was obtained by summing the annual utility-related expenses and adding 10% (10-year payback period) of the installed equipment cost. Detailed calculations based on the research of Liu *et al.* [37] and Douglas' cost correlations [38] were applied according to equation (2.1).

$$\text{TAC} = \frac{\text{capacity cost}}{\text{payback period}} + \text{operating cost}. \tag{2.1}$$

Aspen Plus simulator was used to design the biomass polyol separation model. The conventional distillation sequence and DWC model used in the simulation are shown in figure 2*a* and 2*b*, respectively. Figure 2*b* shows the equivalent four-column model of the simulated DWC, and N1, N2, N3 and N4 represent the tray numbers of the top, prefractionator, side and bottom parts of the DWC. The conditions of the feed mixture and product requirements are listed in table 2.

## 2.2. Optimization

### 2.2.1. Response surface methodology and Box–Behnken design

Box–Behnken design (BBD), which has been widely used in optimizing chemical industrial processes [39–41], was employed under RSM to study the interactions between variables and optimize the system to achieve maximum TAC savings. The comparison [42,43] between the BBD and other response surface designs (central composite, Doehlert matrix and three-level full factorial design) has demonstrated that BBD was more efficient than other methods especially when the factor number was higher than two and could avoid experiments being performed under extreme conditions where

**Table 2.** Feed condition and product requirement.

| properties | | stream name | | | |
|---|---|---|---|---|---|
| pressure | 15 kPa | feed | D | S | W |
| composition (mass) | 1,2-PG | 0.3 | 0.99 | — | — |
| | EG | 0.4 | — | 0.99 | — |
| | 1,3-PG | 0.3 | — | — | 0.99 |
| average volatility | 1,2-PG | 2.44 | | | |
| | EG | 1.78 | | | |
| | 1,3-PG | 1 | | | |
| flow rate (kmol h$^{-1}$) | | 100 | 30 | 40 | 30 |

unsatisfactory results might occur. The BBD only had three levels (low, medium and high, coded as −1, 0, and +1, respectively) and required a few simulations or experimental runs to determine the possible inter-parameter effects on TAC savings in this research. Equation (2.2) is the nonlinear regression model used to fit the simulated data considering all the linear terms, square terms and linear-by-linear interaction terms [44].

$$y = \beta_0 + \sum_{i=1}^{k} \beta_i x_i + \sum_{i=1}^{k} \beta_{ii} x_i^2 + \sum \sum_{i<j} \beta_{ij} x_i x_j + \varepsilon, \tag{2.2}$$

where $\beta_0$ is a constant, $\beta_i$ is the slope or linear effect of input factor $x_i$, $\beta_{ij}$ is the linear-by-linear interaction effect among the input factors $x_i$ and $x_j$, $\beta_{ii}$ is the quadratic effect of input factor $x_i$, and $\varepsilon$ is the error term. MINITAB software was employed to fit the response surface and optimize the TAC savings.

## 2.2.2. Variable analysis

Most reported studies have focused on the optimization of the tray number or reflux ratio of DWCs, while other variables, including the thermal feed states ($q_F$), locations of the feed and side stream, and split ratios of liquid ($S_L$) and gas ($S_G$), have often been ignored by researchers [13,45,46]. However, a suitable feed position would increase the efficiency of separation and avoid a material remixing effect, and different feed $q_F$ values have a large effect on the proportion of gas–liquid phases and mass transfer, especially in the prefractionator. Therefore, both feed location and $q_F$ value influenced the composition of the streams in the top and bottom of the prefractionator, thereby affecting the concentration distribution in the connecting side column and the purity of the side stream. In addition, the side stream should be located at the position with the highest concentration of side product and is affected by the gas–liquid and composition distribution of the side column. As a result, the $q_F$ value and locations of the feed and side products should be simultaneously optimized. The locations of the feed and side stream can be represented as the ratio of the feed tray number ($r_F$) and side stream tray number ($r_S$), respectively, to the total trays in the prefractionator and side column. Larger ratios mean lower locations of the streams. These ratios can more clearly display the relative location and consider changes in total trays.

The adjustment and control of vapour splits was often a challenge in design and the operation of DWCs. Fortunately, recent studies [20,47,48] showed considerable progress in the design of vapour splitter and effective control for vapour split in DWCs, some of which were very promising to be put into practical industrial use [47,48]. On the other hand, Rangaiah *et al*. [49] reported that the liquid and vapour splits had an important effect on the energy consumption of DWCs, so the optimization of liquid and vapour splits was achievable and necessary in the process of DWCs. As far as we know, varied reflux ratio ($R$) values correspond to different gas–liquid proportions and determine product purity to some extent. Similarly, the liquid and gas split ratios affect gas–liquid mass transfer by varying the liquid–gas proportions in the prefractionator and side column. As a result, this so-called 'pseudo reflux ratio' in the prefractionator and side column will greatly influence product purity and the total cost of the DWC and should be studied comprehensively during optimization.

**Table 3.** Simulation results for the conventional distillation sequence and DWC.

| parameter | conventional column | | DWCs | | | |
|---|---|---|---|---|---|---|
| column | T1 | T2 | N1 | N2 | N3 | N4 |
| tray number | 54 | 61 | 26 | 38 | 41 | 9 |
| feed location | 29 | 37 | 17 | | | |
| side stream location | — | — | 21 | | | |
| reflux ratio | 10.5 | 9.3 | 16.4 | | | |
| liquid split | — | — | 0.33 | | | |
| gas split | — | — | 0.41 | | | |
| reboiler duty (GJ h$^{-1}$) | 19.36 | 25.71 | 29.98 | | | |
| condenser duty (GJ h$^{-1}$) | 19.01 | 25.62 | 29.85 | | | |
| TAC ($) | 2 500 300 | | 1 620 195 | | | |
| TAC savings (%) | — | | 35.2 | | | |

# 3. Results and discussions

## 3.1. Design of the conventional columns and divided wall columns

A conventional two-column distillation sequence was designed from the conceptual stage to rigorous simulation through Aspen Plus, and the feed locations, tray numbers and reflux ratio were then optimized based on the minimum TAC. A short-cut design of the DWC was developed based on the FUGK model according to reported research [50,51]. The results of both designs are shown in table 3, and the TAC savings were calculated according to equation (3.1).

$$\text{Savings} = \frac{\text{TAC}_C - \text{TAC}_D}{\text{TAC}_C} \times 100, \tag{3.1}$$

where $\text{TAC}_C$ and $\text{TAC}_D$ are the TACs of conventional distillation and the DWC, respectively. From table 3, the DWC could save 35.2% of the TAC with a lower energy loading and number of trays compared to the conventional distillation sequence.

## 3.2. Optimization of the divided wall columns

Table 4 shows the relative factors and levels of whole variables in the biopolyol-separation DWC. The values of these variables were chosen to fall within a reasonable range based on the initial short-cut design to make the simulation converge. For each run of studied variables, other variables were varied to minimize the TAC in the process of obtaining a quality product. The quadratic model of the response is summarized in electronic supplementary material, table S2, and the low $p$ value (less than 0.005) and high $R^2$ value (greater than 0.9) indicated that these models fit the simulation runs well [52].

### 3.2.1. Optimization of $q_F$ and the locations of feed and side products

Interactions between the thermal feed states and the locations of the feed and side stream were investigated through RSM, and the results are shown in figure 3. From figure 3a,b, an extremely high location of the feed negatively affected the TAC savings, which may be because the composition of the feed was considerably different from that at high locations. Figure 3c indicates that the effect of different combinations of $q_F$ and $r_S$ on TAC was not significant, while middle values were good for TAC savings.

The optimization plot in figure 4 gives the greatest TAC savings (39.64%) according to the variables of $q_F$, $r_F$ and $r_S$, which had coded values of 0.72, 0.182 and −0.11, respectively. In addition, feed location ($r_F$) had a significant effect on the TAC savings of the DWC compared with that of the other two variables, and a higher $r_F$ would minimize TAC. A middle location of the side column and near-saturated liquid feed would be the optimum conditions for the biopolyol DWC.

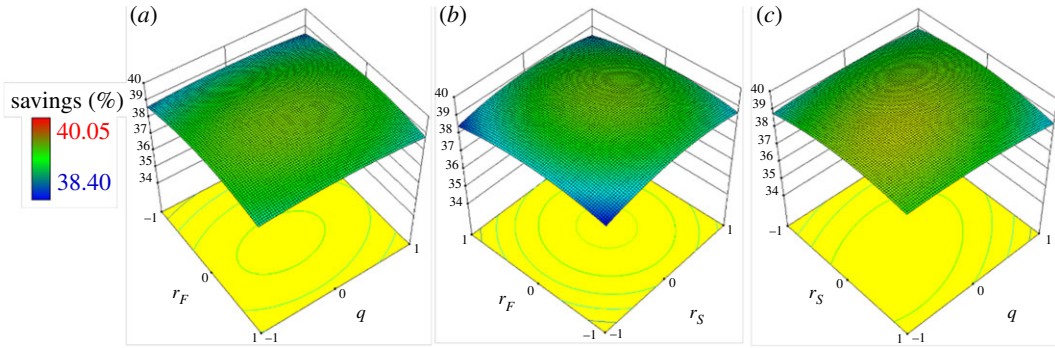

**Figure 3.** Three-dimensional response surface plots between the variables $q_F$, $r_F$ and $r_S$.

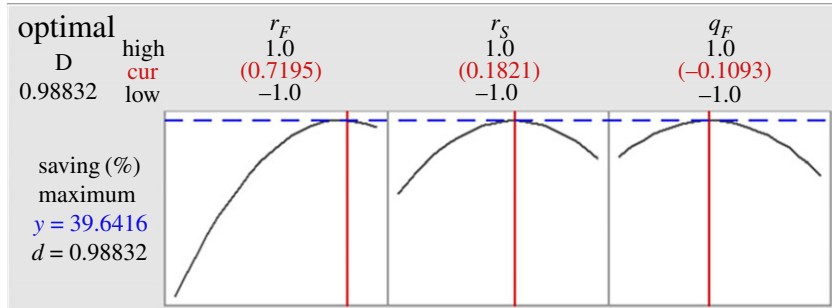

**Figure 4.** Optimization plot for $q_F$, $r_F$ and $r_S$ by BBD.

**Table 4.** Levels of variables for BBD.

| factors | levels | | |
|---|---|---|---|
| code value | −1 | 0 | 1 |
| thermal feed states ($q_F$) | −0.5 | 0 | 0.5 |
| feed location ($r_F$) | 0.4 | 0.45 | 0.5 |
| side stream location ($r_S$) | 0.47 | 0.52 | 0.57 |
| top section tray (N1) | 21 | 26 | 31 |
| prefractionator tray (N2) | 31 | 38 | 45 |
| side column tray (N3) | 33 | 41 | 49 |
| bottom section tray (N4) | 7 | 9 | 11 |
| liquid split ($S_L$) | 0.3 | 0.33 | 0.36 |
| gas split ($S_G$) | 0.38 | 0.43 | 0.47 |
| reflux ratio ($R$) | 15.6 | 17.4 | 19.1 |

### 3.2.2. Optimization of N1–N4

Interactions between the trays in the four parts of the DWC (N1–N4) were investigated through RSM, and the results are shown in figure 5. The interaction between N2 and N3 had the greatest effect on TAC savings, while the N1–N4 interaction affected TAC to a lesser extent. In each interaction of terms, N2 and N3 were obvious key factors for TAC savings, which may be due to the middle column being the main separation zone between the intermediate component (EG) and light component (1,2-PG) or heavy component (1,3-PG).

The optimization plot in figure 6 gives the maximum TAC savings (40.31%) according to the variables N1–N4, which had values of −0.35, 0.62, 0.74 and −0.1, respectively. Moreover, TAC decreased with increasing N2 and N3, and higher savings were achieved with high tray heights in the middle column. As a result, a relatively large number of trays in the prefractionator and side column would

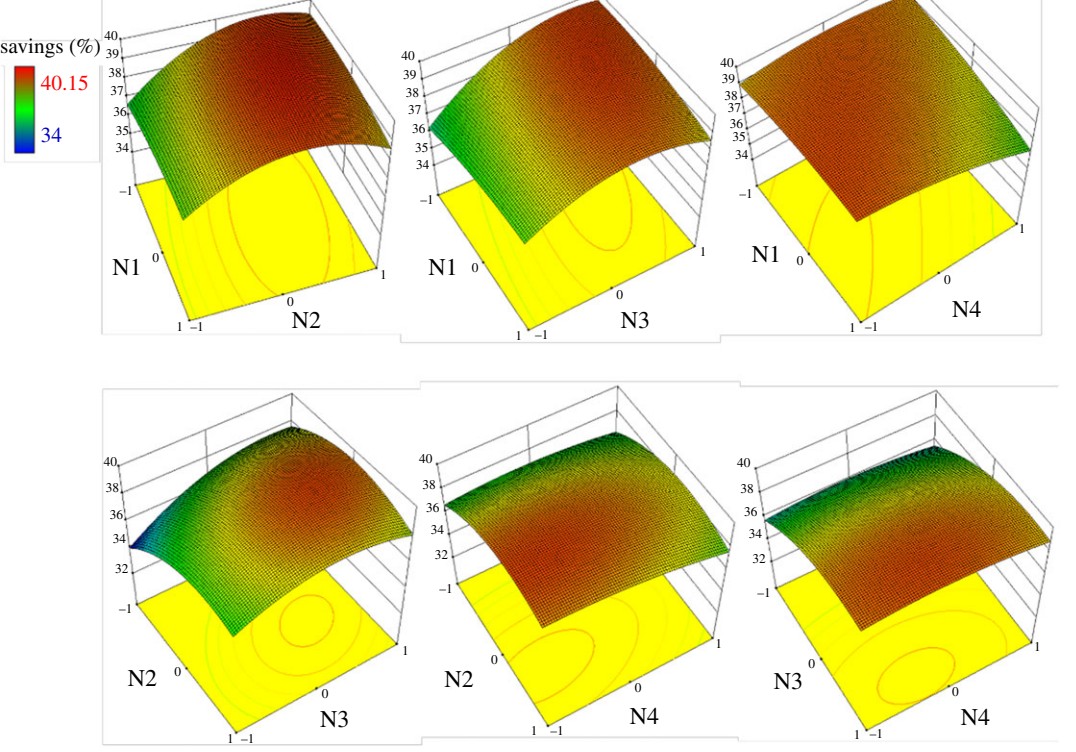

**Figure 5.** Three-dimensional response surface plots between the variables N1–N4.

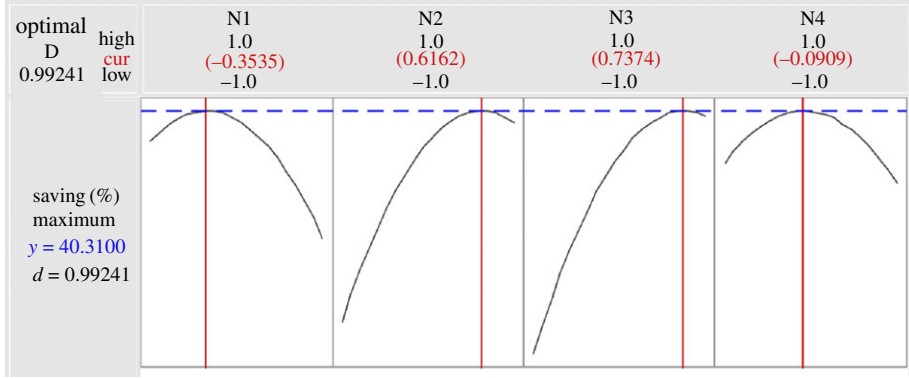

**Figure 6.** Optimization plot for N1–N4 by BBD.

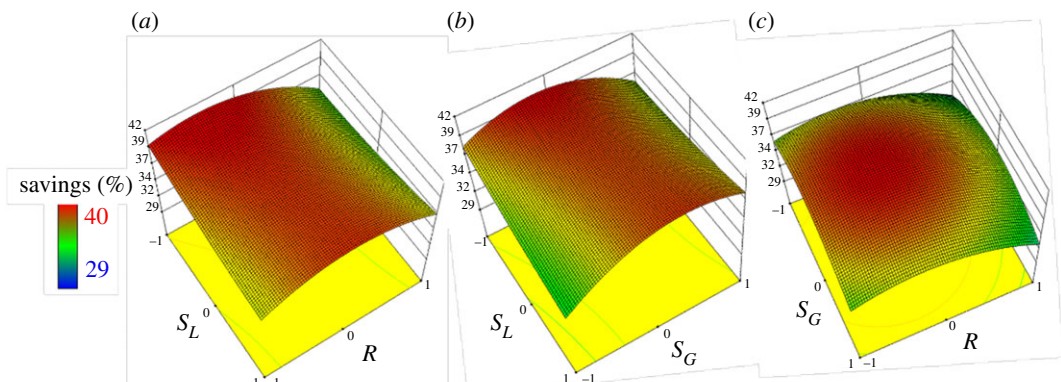

**Figure 7.** Three-dimensional response surface plots between the variables $S_L$, $S_G$ and $R$.

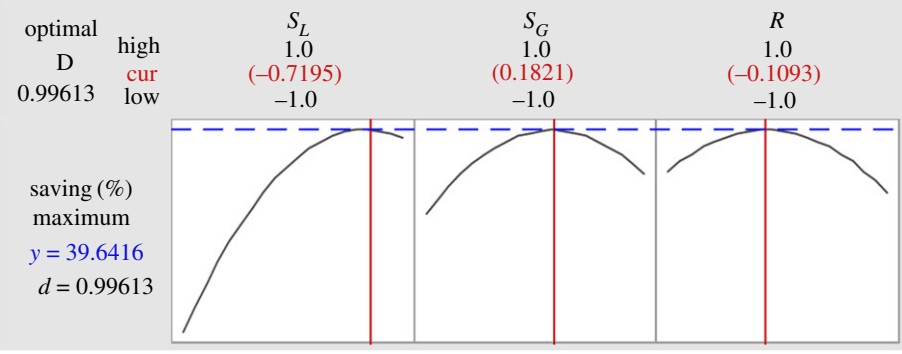

**Figure 8.** Optimization plot for $S_L$, $S_G$ and $R$ by BBD.

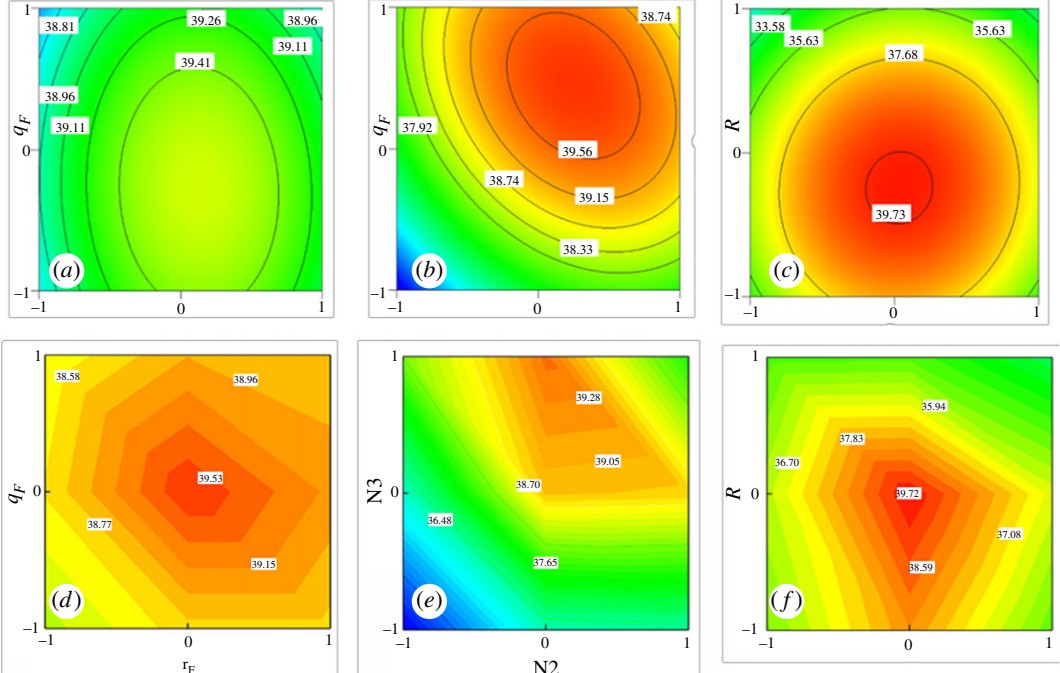

**Figure 9.** Contour plots of interactions predicted by BBD ($a$–$c$) and by the simulation ($d$–$f$).

lower the energy consumption. However, as the diameter of the top and bottom parts in the DWC was the sum of the diameters of the middle columns, more trays at both ends would increase the capital investment.

### 3.2.3. Optimization of liquid split ratio, gas split ratio and reflux ratio

As discussed above, the split ratios of liquid and gas and reflux ratio heavily influence gas–liquid mass transfer in the middle part of the DWC. Therefore, interactions between $S_L$, $S_G$ and $R$ (figure 7) showed the greatest effect on TAC savings through the response results. In particular, figure 7$b$ shows that a suitable $S_L$, selected to match $S_G$, was necessary to save costs for the biopolyol DWC. In addition, under the premise of minimum $R$, a relatively low reflux ratio, which means a low energy load, would greatly benefit the DWC (figure 7$a$,$c$).

As shown in figure 8, optimized $S_L$ (0.72), $S_G$ (0.18) and $R$ (−0.11) values resulted in TAC savings reaching 39.64%. Above all, an unsuitable $S_L$ would greatly reduce the savings for the DWC, which may be because the feed and side stream are liquids and an improper distribution of liquid would adversely affect gas–liquid contact in the column. An uneven proportion of gas–liquid in the middle part of the column can be called a bad 'pseudo $R$'. However, an extremely low reflux ratio in the DWC would burden capital investment, while an excessively high value of $R$ would require excessive

**Table 5.** Comparison of variables among the short-cut design, RSM and simulation.

| variables | short-cut design | RSM | simulation |
|---|---|---|---|
| $q_F$ | 0 | −0.055 | −0.055 |
| $r_F$ | 0.447 | 0.486 | 0.486 |
| $r_S$ | 0.51 | 0.53 | 0.53 |
| N1 | 32 | 24 | 25 |
| N2 | 38 | 42 | 42 |
| N3 | 41 | 47 | 46 |
| N4 | 9 | 9 | 9 |
| $S_L$ | 0.33 | 0.35 | 0.35 |
| $S_G$ | 0.43 | 0.41 | 0.41 |
| R | 17.4 | 16.2 | 16.05 |
| TAC ($) | 1 620 195 | 1 492 429 | 1 472 926 |
| TAC savings (%) | 35.2 | 40.31 | 41.09 |

energy consumption. As a result, a relatively low reflux ratio is appropriate for saving costs for the biopolyol DWC and ensuring product quality.

## 3.3. Validation of the optimization

Several simulations were carried out to validate the fitted models from the RSM, and figure 9 shows the comparison of results between BBD and rigorous runs. The RSM plots fit those of the actual simulation. The final simulation was run according to the optimized values of all variables in the biopolyol DWC, and the obtained results are compared with the short-cut design and RSM predicted values in table 5.

The optimized results from RSM were used as initial values for simulation with minor adjustments for convergence and product quality. From table 5, the actual ultimate values of the variables were close to those predicted by RSM. The TAC of the biopolyol DWC could be reduced to $1 472 926, and the expense was reduced by approximately 6% compared with that of the short-cut design. However, it could be concluded from a comparison between the short-cut design and final optimization that it was necessary to increase the tray numbers of the prefractionator and side column with optimal split ratios of liquid and gas. These adjustments would lower the number of trays in the top column and the reflux ratio, thus reducing the investment and process cost. In addition, the purity (greater than 99 mol%) and yield of the corresponding streams (electronic supplementary material, table S3) in the optimized DWC achieved the design requirements.

## 4. Conclusion

In this work, a DWC for separating near-volatile biomass polyols was designed, and a conventional distillation sequence was set as the reference for economic evaluation. Then, a practical method was proposed for the economic optimization of the DWC based on RSM. The optimization of whole variables was efficiently carried out with BBD with the aim of maximizing TAC savings. The effects and interactions between all factors were studied and discussed. The results showed that more trays in the prefractionator and side column would be beneficial for TAC savings. Moreover, $S_L$ and $S_G$, which can be called the 'pseudo reflux ratio', were the most influential factors on the TAC of DWC. Optimal values of these significant variables can reduce the capital investment and energy consumption. Finally, the variable values predicted by RSM showed good agreement with those of the actual simulation, and the final optimized biopolyol DWC reduced the TAC by 41.09% compared with conventional distillation. As it is more expensive to separate near-volatile biopolyol than an easily separated system, the optimization process could save considerable expense. Cost savings in the separation of near-volatile biomass polyol would promote the development of green biomass and improve its economic value. This paper demonstrated that RSM is a powerful and reliable technique for the design and optimization of complex DWCs.

# Abbreviations

1,2-PG: 1,2-propanediol
1,3-PG: 1,3-propanediol
ANOVA: analysis of variance
BBD: Box–Behnken design
DWC: divided wall column
EG: ethylene glycol
MINLP: mixed integer nonlinear programming
MILP: mixed integer linear programming
RSM: response surface methodology
TAC: total annual cost

Data accessibility. Physical-chemical data of polyol including binary interactive parameters of NRTL model, relative volatility and VLE data of polyol were shown in Table S1, Fig S1 and Fig S2 respectively. Table S2 described the calculated second-order model for DWCs' variables from RSM and the results of streams including composition and flowrate in optimized DWCs were shown as Table S3.

Authors' contributions. T.C. designed the study, carried out the simulation work and drafted the manuscript; L.L. carried out the statistical analyses and participated in the simulation work; Y.C. carried out the calculation of obtained data and helped draft the manuscript; P.B. conceived of the study and coordinated the study. All authors gave final approval for publication.

Competing interest. We declare we have no competing interests.

Funding. We received no funding for this study.

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
