## [Reviewer comments · Royal Society Open Science]

Review History

RSOS-191748.R0 (Original submission)

Review form: Reviewer 1

Is the manuscript scientifically sound in its present form?

Yes

Are the interpretations and conclusions justified by the results?

Yes

Is the language acceptable?

Yes

Do you have any ethical concerns with this paper?

No

Have you any concerns about statistical analyses in this paper?

No

Recommendation?

Major revision is needed (please make suggestions in comments)

Comments to the Author(s)

This problem is relevant for journal scope. The topic of the article is up to date, the introduction and literature survey is easy to understand and detailed, the authors discussed already all available literature sources. The modelling approach is very professional. The presentation and discussion of the result is clear and professional. The conclusions are well extracted from the results and discussion.

I suggest the acceptance after major revision. The concept and aim are clearly defined. I could not find typing errors.

Remarks, suggestions, questions

1. Please cite more papers from this journal at the last two years in the similar topic of this research.
2. Please distinguish the theory, introduction part from practice. I suggest the following the standard article structure: Introduction, Material and Methods, Results and Discussion, Conclusions.
3. Please introduce the physical-chemical parameters of the separated materials: EG, 1,2-PG, 1,3-PG.
4. Please introduce the vapour-liquid equilibrium of the separated mixture.
5. Did you make the optimisation of the (feed) pressure too?
6. Table 2: The reflux ratio has no unit. Reflux stream has a unit, e. g. mole.
7. Table 2: Please use J or MJ, instead Gcal.

Review form: Reviewer 2 (Chunjian Xu)

Is the manuscript scientifically sound in its present form?

Yes

Are the interpretations and conclusions justified by the results?

Yes

Is the language acceptable?

Yes

Do you have any ethical concerns with this paper?

No

Have you any concerns about statistical analyses in this paper?

No

Recommendation?

Accept with minor revision (please list in comments)

Comments to the Author(s)

Please find the comments as attached (Appendix A).

Decision letter (RSOS-191748.R0)

14-Feb-2020

Dear Dr Tao,

The editors assigned to your paper ("A comprehensive economic optimization methodology of divided wall columns for biopolyol separation") have now received comments from reviewers. We would like you to revise your paper in accordance with the referee and Associate Editor suggestions which can be found below (not including confidential reports to the Editor). Please note this decision does not guarantee eventual acceptance.

Please submit a copy of your revised paper before 08-Mar-2020. Please note that the revision deadline will expire at 00.00am on this date. If we do not hear from you within this time then it will be assumed that the paper has been withdrawn. In exceptional circumstances, extensions may be possible if agreed with the Editorial Office in advance. We do not allow multiple rounds of revision so we urge you to make every effort to fully address all of the comments at this stage. If deemed necessary by the Editors, your manuscript will be sent back to one or more of the original reviewers for assessment. If the original reviewers are not available, we may invite new reviewers.

- Data accessibility

If you wish to submit your supporting data or code to Dryad (<http://datadryad.org/>), or modify your current submission to dryad, please use the following link:
<http://datadryad.org/submit?journalID=RSOS&manu=RSOS-191748>

- **Competing interests**

- **Authors' contributions**

- **Acknowledgements**

- **Funding statement**

on behalf of Professor Hazel Assender (Associate Editor) and R. Kerry Rowe (Subject Editor)
openscience@royalsociety.org

Reviewers' Comments to Author:

Reviewer: 1

Comments to the Author(s)

This problem is relevant for journal scope. The topic of the article is up to date, the introduction and literature survey is easy to understand and detailed, the authors discussed already all available literature sources. The modelling approach is very professional. The presentation and

discussion of the result is clear and professional. The conclusions are well extracted from the results and discussion.

I suggest the acceptance after major revision. The concept and aim are clearly defined. I could not find typing errors.

Remarks, suggestions, questions

1. Please cite more papers from this journal at the last two years in the similar topic of this research.
2. Please distinguish the theory, introduction part from practice. I suggest the following the standard article structure: Introduction, Material and Methods, Results and Discussion, Conclusions.
3. Please introduce the physical-chemical parameters of the separated materials: EG, 1,2-PG, 1,3-PG.
4. Please introduce the vapour-liquid equilibrium of the separated mixture.
5. Did you make the optimisation of the (feed) pressure too?
6. Table 2: The reflux ratio has no unit. Reflux stream has a unit, e. g. mole.
7. Table 2: Please use J or MJ, instead Gcal.

Reviewer: 2

Comments to the Author(s)

Please find the comments as attached

Author's Response to Decision Letter for (RSOS-191748.R0)

See Appendix B.

Decision letter (RSOS-191748.R1)

11-Mar-2020

Dear Dr Tao,

It is a pleasure to accept your manuscript entitled "A comprehensive economic optimization methodology of divided wall columns for biopolyol separation" in its current form for publication in Royal Society Open Science.

Please ensure that you send to the editorial office an editable version of your accepted manuscript, and individual files for each figure and table included in your manuscript. You can

send these in a zip folder if more convenient. Failure to provide these files may delay the processing of your proof. You may disregard this request if you have already provided these files to the editorial office.

on behalf of Professor Hazel Assender (Associate Editor) and R. Kerry Rowe (Subject Editor)
openscience@royalsociety.org

Appendix A

In the manuscript entitled “A comprehensive economic optimization methodology of divided wall columns for biopolyol separation”, the authors have carefully investigated the separation of biopolyols in the DWCs, and the RSM method was employed to optimize this strong non-linear process. This topic is of significant importance for the chemical industry and the manuscript is well organized. Thus, I suggest the publication of this work after the following issues being addressed.

1. In the manuscript, the feed composition was (1,2-PG, EG, 1,3-PG)=(0.3, 0.4, 0.3), is this composition originated from the real chemical industry or open literature? It would be much better to explain the reasons for choosing this composition for the case study.
2. The phys-chemical properties of each component, such as boiling points, should be mentioned in the text or tables to help readers to identify the feasible sequences quickly.
3. As stated in the manuscript, the interactions between different variables are of key importance for the overall optimization of the DWC process, why the Box-Behnken design (BBD) was employed to investigate the interactions? Is there any other design having better optimization performance?
4. Liquid and vapor split ratios (S_L and S_G) are the most important optimization factors, and this conclusion is consistent with the previous studies. From the process control aspect, the adjustment of S_L can be easily achieved, however it would be very difficult to adjust S_G . Thus, are there any other feasible optimization couples with S_G fixed?

Appendix B

Manuscript ID: RSOS-191748

Title: “A comprehensive economic optimization methodology of divided wall columns for biopolyol separation”

Author(s): Tao Chen, Lingjuan Lv, Yuanzhi Chen, Peng Bai

Dear Lianne Parkhouse, Prof. Hazel Assender and R. Kerry Rowe,

Thank you for handling our manuscript. We received the reviews for the aforementioned manuscript, and we are submitting a revised version of the manuscript (“*Manuscript*”) and supporting materials. First, a “*Supporting Information*” (SI) document including some supporting data was added in this submission. According to reviewers’ advice, we added two figures and three tables into this SI document where Table S2 and Table S3 were from aforementioned manuscript as Table 4 and Table 6. The content of SI was described below the “*Conclusion*” section, and all data in SI was cited and explained in manuscript. Secondly, Author’s contribution was rewritten in a suggested form below the “*Competing Interests*”. Moreover, the changes made to the original manuscript are kept the trace of modification in the document called “*Manuscript with revision marks*” and the responses to specific comments of the reviewers are provided in this letter. Itemized response to reviewers’ comments is appended below.

Sincerely,

Tao Chen

Reviewer(s)' Comments to Author:

Reviewer#1

General Comments:

This problem is relevant for journal scope. The topic of the article is up to date, the introduction and literature survey is easy to understand and detailed, the authors discussed already all available literature sources. The modelling approach is very professional. The presentation and discussion of the result is clear and professional. The conclusions are well extracted from the results and discussion.

I suggest the acceptance after major revision. The concept and aim are clearly defined.

I could not find typing errors

Reply: Thank you sincerely for your constructive comments and valuable recommendations. We have carefully revised the manuscript according to your suggestions.

Q 1. Please cite more papers from this journal at the last two years in the similar topic of this research.

Answer: Thanks for your careful review and this advice would complete the Introduction section. We have cited more recent articles about design and optimization of DWCs. Related description of these articles was added in the second, third and fourth paragraph of Introduction section as follows:

“Many researches about the short-cut design of DWCs were based on the Fenske-Underwood-Gilliland-Kirkbride (FUGK) method aimed at minimum vapor flow rate¹⁻³. Besides, recent articles have reported other systematic methods and models for divided-wall column design: a rigorous model validated in the experiment was used for the scaled design⁴; coordinate descent methodology with a random research⁵ and optimization-based design approach with an automated initialization procedure were employed to determine variables of DWCs during the initial design⁶. As a result, not only the short-cut design but further optimization for this complex chemical units

should be attached great importance for separation mission, and different values of these variables could have an unequal effect on the energy or capital costs of the separation process.”

“Gómez-Castro et al.⁷ proposed an optimization technique using genetic algorithms for an alternative design with a post-fractionator instead of a prefractionator, and multi-objective genetic algorithms was also used to optimize heat pump assisted reactive dividing wall column considering the economic and thermodynamic efficiency performances⁸. To handle disturbance and implementation error of DWCs, Luo et. al⁹ developed an online optimization method assisted with steady-state analysis. In addition, a sequential quadratic method using the mixed integer linear¹⁰ or nonlinear¹¹ programming problem (MINLP) approach was also used for conceptual design of DWCs.”

“As a result, RSM is generally used in optimization research on multivariable systems, including the simulation of structural or process parameters in DWCs^{12, 13}.”

Relevant literatures have been added to the reference part.

Q 2. Please distinguish the theory, introduction part from practice. I suggest the following the standard article structure: Introduction, Material and Methods, Results and Discussion, Conclusions.

Answer: *According to your suggestion, the structure of manuscript was reconstructed based on the classification of contents. As a result, the manuscript was divided into four sections as follows:*

- 1. Introduction.*
- 2. Design and Optimization Methods: Describing the properties of the studied mixtures, methods and theory used in design and optimization.*
- 3. Results and Discussions: Showing the results of design and optimization for distillation columns, discussing the interactions between optimized variables and their effect on the total annual cost, validating the reliability of optimization methodology and the obtained results.*
- 4. Conclusions.*

In detail, the results of initial design of distillation in section 2.1 was moved to 3.1,

and optimization theory and method in section 3 (3.1 and 3.2) was moved to section 2.2 as 2.2.1 and 2.2.2 respectively.

Your valuable advice makes the structure of this manuscript more reasonable and is of great importance for this research. Thank you.

Q 3. Please introduce the physical-chemical parameters of the separated materials: EG, 1,2-PG, 1,3-PG.

Answer: Thanks for your comments. Related parameters of the separated polyol, including the binary interactive parameters of NRTL model and their boiling points at operation pressure, were added as Table S1 in Supporting Information (SI) and the first paragraph of section 2.1 in manuscript.

Table S1 Binary interactive parameters of NRTL model

Component i	Component j	A_{ij}	A_{ji}	B_{ij}/K	B_{ji}/K	C_{ij}
1,2-PG	EG	-1.28	-0.43	995.38	-136.44	0.30
1,2-PG	1,3-PG	-1.54	-0.65	465.27	636.46	0.50
EG	1,3-PG	-2.95	1.74	1002.23	-246.97	0.27

Related references and corresponding descriptions were added in manuscript as follows:

“Physical-chemical properties for polyol were taken from the Aspen Plus component database. The NRTL¹⁴ model was chosen to describe the phase behaviour of this polyol system and all binary interactive parameters from Aspen Plus were listed in Table S1 in Supporting Information and showed good consistency with reports¹⁵⁻¹⁷. Moreover, polyol has relatively high boiling point (1,2-PG: 461.3 K; EG: 470.8 K; 1,3-PG: 483.7 K) at 1 atm.... the boiling point of 1,2-PG, EG, 1,3-PG was 406.3 K, 415.2 K and 431.7 K respectively at operation pressure.”

Q 4. Please introduce the vapor-liquid equilibrium of the separated mixture.

Answer: Vapor-liquid equilibrium of separated mixtures could help us identify the feasible sequences quickly and clear. According to your advice, we added the VLE data

at operation pressure graphically in SI document as Fig S2, related description and references was added at the end of first paragraph of section 2.1 in manuscript as follows:

Fig S2. VLE data for (A) 1,2-PG + EG, (B) EG + 1,3-PG, (C) 1,2-PG + 1,3-PG at 15 kPa.

“Moreover, the vapor-liquid equilibrium data of polyol¹⁵⁻¹⁷ was presented graphically in Fig S2 which indicated that the boiling point of 1,2-PG, EG, 1,3-PG was 406.3 K, 415.2 K and 431.7 K respectively at 15 kPa.”

We appreciate very much for your suggestion which could make this study more rigorous and scientific.

Q 5. Did you make the optimization of the (feed) pressure too?

Answer: Thank you for your comment. As this research was about the design of the practical industrial project, we mainly took the temperature of heat source and the relative volatility of the polyol into account before design. Polyol has relatively high boiling point (1,2-PG: 461.3 K; EG: 470.8 K; 1,3-PG: 483.7 K) at 1 atm, which means

that high-grade heat source may be needed in the reboiler. So, vacuum distillation was necessary for the sake of cost saving. However, as shown in Fig. S1, when the liquid mole fraction of EG is lower than 0.2 around, the relative volatility of polyol decreases as pressure dropping from 100 kPa to 5 kPa and a relatively large drop was found from 15 kPa to 5 kPa. As a result, to avoid harsh requirement for heat source and terrible separation efficiency, 15 kPa was chosen as operation pressure in this study.

According to your comment, Fig S1 was added in SI document and related explanation and description was added in the middle of first paragraph of section 2.1 in manuscript as follows:

Fig S1. Relative volatility of (A) 1,2-PG/EG, (B) EG/1,3-PG at different pressures.

“Moreover, polyol has relatively high boiling point (1,2-PG: 461.3 K; EG: 470.8 K; 1,3-PG: 483.7 K) at 1 atm and its boiling need expensive high-grade heat source during distillation process. In another hand, as shown in Fig S1, when the liquid mole fraction of EG is lower than 0.2 around, the relative volatility of 1,2-PG/EG and EG/1,3-PG were lower at vacuum condition than that at high pressure, and this decline becomes relatively heavy from 15 kPa to 5 kPa, which is bad for polyol’s separation. As a result, considering both relatively lower heat requirement and higher separation efficiency, 15 kPa was chosen as the operation and feed pressure of distillation unit.”

Your comment on this aspect is significant and makes this study rigorous. Thank you.

Q 6. Table 2: The reflux ratio has no unit. Reflux stream has a unit, e. g. mole.

Answer: Thank you for your careful review. We have removed the unit of reflux ratio in Table 2.

Q 7. Table 2: Please use J or MJ, instead Gcal.

Answer: Thank you for your recommendation. Indeed, J, MJ and GJ are international system of units generally used in published researches. According to your suggestion, we have replaced all units of heat duty from Kcal/h to GJ/h considering their magnitude, and related values have also been revised according to their units in Table 2.

At last, we want to thank you sincerely for your significant suggestions again, these valuable and professional advice will greatly improve the quality of this paper. And I feel grateful that so much of your precious time was spent on our paper revision.

Reviewer#2

General Comments: In the manuscript entitled “A comprehensive economic optimization methodology of divided wall columns for biopolyol separation”, the authors have carefully investigated the separation of biopolyols in the DWCs, and the RSM method was employed to optimize this strong non-linear process. This topic is of significant importance for the chemical industry and the manuscript is well organized. Thus, I suggest the publication of this work after the following issued being addressed.

Reply: Thank you sincerely for your constructive comments and valuable recommendations. We have carefully revised the manuscript according to your suggestions. The itemized response is attached as follows:

Q 1. In the manuscript, the feed composition was (1,2-PG, EG, 1,3-PG) = (0.3, 0.4, 0.3), is this composition originated from the real chemical industry or open literature? It would be much better to explain the reasons for choosing this composition for the case study.

Answer: *Thank you for your advice. The composition of polyol generated from glycerol hydrogenolysis is mainly decided by the catalyst types and reaction conditions¹⁸⁻²¹. The design and optimization of distillation units in this study was for the practical industrial engineering based on industrially valuable biomass conversion using Cu/Ni-based catalyst²¹⁻²³ with relatively high conversion rate of glycerol. However, it is difficult to separate three components with close quantity through distillation. So, considering the high industrial value and the necessity of separation, we chose 1,2-PG (~ 30 mole %), EG (~ 40 mole %) and 1,3-PG (~ 30 mole %) as a case to be studied in this research. According to your suggestion, Corresponding explanation was added and related references was cited in the beginning of section 2.1 of this manuscript as follows:*

“The conversion rate of glycerol and the composition of generated three polyol in hydrogenolysis reaction mainly depend on the catalyst and reaction conditions¹⁸⁻²¹. In this research, the separation study was based on the product from the industrially valuable biomass conversion using Cu/Ni-based catalyst²¹⁻²³ with high conversion rate

of glycerol and the composition of main biopolyol products from glycerol hydrogenolysis was chosen to be 1,2-PG (~ 30 mole %), EG (~ 40 mole %) and 1,3-PG (~ 30 mole %), because it was challenging but meaningful to obtain three purified products with close quantity through distillation.”

We appreciated very much for your valuable advice which could make this research more meaningful.

Q 2. The phys-chemical properties of each component, such as boiling points, should be mentioned in the text or tables to help readers to identify the feasible sequences quickly.

Answer: Thanks for your comments. Related parameters of the separated polyol, including the binary interactive parameters of NRTL model, vapor-liquid equilibrium data and their boiling points at operation pressure, were added as Table S1, Fig S2 in Supporting Information (SI) and the first paragraph of section 2.1 in manuscript respectively. Related references and corresponding descriptions were added in manuscript as follows:

Table S1 Binary interactive parameters of NRTL model

Component i	Component j	A_{ij}	A_{ji}	B_{ij}/K	B_{ji}/K	C_{ij}
1,2-PG	EG	-1.28	-0.43	995.38	-136.44	0.30
1,2-PG	1,3-PG	-1.54	-0.65	465.27	636.46	0.50
EG	1,3-PG	-2.95	1.74	1002.23	-246.97	0.27

Fig S2. VLE data for (A) 1,2-PG + EG, (B) EG + 1,3-PG, (C) 1,2-PG + 1,3-PG at 15 kPa.

“Physical-chemical properties for polyol were taken from the Aspen Plus component database. The NRTL¹⁴ model was chosen to describe the phase behaviour of this polyol system and all binary interactive parameters from Aspen Plus were listed in Table S1 in Supporting Information and showed good consistency with reports¹⁵⁻¹⁷.... Moreover, the vapor-liquid equilibrium data of polyol¹⁵⁻¹⁷ was presented graphically in Fig S2 which indicated that the boiling point of 1,2-PG, EG, 1,3-PG was 406.3 K, 415.2 K and 431.7 K respectively at operation pressure.”

Q 3. As stated in the manuscript, the interactions between different variables are of key importance for the overall optimization of the DWC process, why the Box-Behnken design (BBD) was employed to investigate the interactions? Is there any other design having better optimization performance?

Answer: *Indeed, several optimization methods in RSM including central composite²⁴, Doehlert matrix²⁵, three-level full factorial²⁶ and Box-Behnken²⁷ design have ever been employed in the design and optimization of chemical or biochemical process. However,*

compared with other designs^{28, 29}, the BBD is a more efficient method and requires less experiments or simulations during optimization. Calculated result from BBD is more reliable and credible because extreme conditions could be precluded through Box-Behnken design. As a result, BBD was employed in our study to optimize the DWCs and discuss the interactions between variables. According to your advice, the characteristics of BBD and the reason we chose it as the method for optimization were added at the beginning of section 2.2.1 as follows:

“Box-Behnken design (BBD), which has been widely used in optimizing chemical industrial processes³⁰⁻³², was employed under RSM to study the interactions between variables and optimize the system to achieve maximum TAC savings. The comparison^{28, 29} between the BBD and other response surface designs (central composite, Doehlert matrix and three-level full factorial design) has demonstrated that BBD was more efficient than other methods especially when the factor number was higher than 2 and could avoid experiments being performed under extreme conditions where unsatisfactory results might occur.”

Thank you for your valuable advice, which could make this study more rigorous.

Q 4. Liquid and vapor split ratios (SL and SG) are the most important optimization factors, and this conclusion is consistent the previous studies. From the process control aspect, the adjustment of SL can be easily achieved, however it would be very difficult to adjust SG. Thus, are there any other feasible optimization couples with SG fixed?

Answer: We totally agree that the control and adjustment of vapor spilt ratios (SG) was relatively difficult in the distillation of divided-wall column process, because the vapor splitter often could not achieve high accuracy and many disturbances may influence the control of SG during the distillation process. However, many researchers have been trying to solve this problems, and recent studies have reported some effective method and equipment such as an enhanced active vapor distributor³³ could achieve the aimed vapor spilt ratio by altering the liquid level of a modified chimney tray, a self-optimizing control methodology³⁴ could stabilize the SG and reject the effects of disturbances through the decentralized control loops. Researches about the vapor spilt control have

made considerable progress and some methods or equipment are industrially valuable in this distillation process³⁴⁻³⁷. As a result, to make comprehensive optimization for polyol's separation, we took the SG into account because its significance for the design of DWCs.

Your comment is very professional and could make our view more reliable in this study. According to your advice, we have added related content about the recent progress in SG control and its significance in DWCs at the second paragraph of section 2.2.2 in manuscript as follows:

“The adjustment and control of vapor splits was often a challenge in design and operation of DWCs. Fortunately, recent studies^{7, 33, 38} showed considerable progress in the design of vapor splitter and effective control for vapor split in DWCs, some of which were very promising to be put into practical industrial^{33, 38}. In another hand, Rangaiah et al.³⁹ reported that the liquid and vapour splits had an important effect on the energy consumption of DWCs, so the optimization of liquid and vapor splits was achievable and necessary in the process of DWCs.”

At last, I want to thank you sincerely for your significant suggestions again, these valuable and professional advice will greatly improve the quality of this paper. And I feel grateful that so much of your precious time was spent on our paper revision.

References:

1. W. Chen, K. Huang, H. Chen, C. Xia, G. Wu and K. Wang, *Chem. Eng. Process.*, 2014, **75**, 90-109.
2. K.T. Chu, L. Cadoret, C.C. Yu and J. D. Ward, *Ind. Eng. Chem. Res.*, 2011, **50**, 9221-9235.
3. Q. H. Ng, S. Sharma and G. P. Rangaiah, *Chem. Eng. Res. Des.*, 2017, **118**, 142-157.
4. M. M. Donahue, M. Baldea and R. Bruce Eldridge, *Chem. Eng. Process.*, 2019, **145**, 107641.
5. N. Van Duc Long, T. N. Pham and M. Lee, *Chem. Eng. Process.*, 2018, **127**, 65-71.
6. T. Waltermann, S. Sibbing and M. Skiborowski, *Chem. Eng. Process.*, 2019, **146**, 107688.
7. C. Molero, A. de Lucas and J. F. Rodríguez, *Solvent Extr. Ion Exch.*, 2006, **24**, 719-730.
8. A. Yang, Y. Su, W. Shen, I. L. Chien and J. Ren, *Energ. Convers. Manage.*, 2019, **199**, 112041.
9. J. Luo, C. Xu, Y. Zhang, K. Yan and J. Zhu, *Comput. Chem. Eng.*, 2018, **119**, 112-127.
10. J. A. Caballero and I. E. Grossmann, *Comput. Chem. Eng.*, 2014, **61**, 118-135.
11. E. Soraya Rawlings, Q. Chen, I. E. Grossmann and J. A. Caballero, *Comput. Chem. Eng.*, 2019, **125**, 31-39.
12. P. B. de Oliveira Filho, M. L. F. Nascimento and K. V. Pontes, *Comput-Aid. Chem. Eng.*, 2018, **43**, 361-366.
13. J. Kaur and V. K. Sangal, *Chem. Eng. Technol.*, 2018, **41**, 1057-1065.
14. H. Renon and J. M. Prausnitz, 1968, *AIChE.*, **14**, 135-144.
15. Nilesh, Vijay, Rane, Alka, Kumari, J., Soujanya, B. and Satyavathi, *Thermochim. Acta*, 2017, **649**, 41-53.
16. L. Zhang, W. Wu, Y. Sun, L. Li, B. Jiang, X. Li, N. Yang and H. Ding, *J. Chem. Eng. Data*, **58**, 1308-1315.
17. C. Yang, F. Xue, Y. Sun, Y. Qian and J. Zhi, *J. Chem. Eng. Data*, 2015, **60**, 1126-1133.
18. J. Ma, W. Yu, M. Wang, X. Jia, F. Lu and J. Xu, *Chinese J. Catal.*, 2013, **34**, 492-507.
19. G. M. Lari, G. Pastore, M. Haus, Y. Ding, S. Papadokostantakis, C. Mondelli and J. Pérez-Ramírez, *Energ. Environ. Sci.*, 2018, **11**, 1012-1029.
20. S. Bagheri, N. M. Julkapli and W. A. Yehye, *Renew. Sust. Energ. Rev.*, 2015, **41**, 113-127.
21. T. Jiang, Q. Huai, T. Geng, W. Ying, T. Xiao and F. Cao, *Biomass Bioenergy*, 2015, **78**, 71-79.
22. E. S. Vasiliadou and A. A. Lemonidou, *Chem. Eng. J.*, 2013, **231**, 103-112.
23. Q. Liu, X. Cao, T. Wang, C. Wang, Q. Zhang and L. Ma, *RSC Advances*, 2015, **5**, 4861-4871.
24. P. A. J. Rosa, A. M. Azevedo and M. R. Aires-Barros, *J. Chromatogr. A*, **1141**, 50-60.
25. S. L. C. Ferreira, W. N. L. d. Santos, C. M. Quintella, B. c. B. Neto and J. M. Bosque-Sendra, *Talanta*, 2004, **63**, 0-1067.
26. S. J. Cheng, J. M. Miao and S. J. Wu, *Renew. Energ.*, 2012, **39**, 250-260.
27. N. Sotudeh and B. Hashemi Shahraki, *Chem. Eng. Technol.*, 2007, **30**, 1284-1291.
28. S. L. C. Ferreira, R. E. Bruns, H. S. Ferreira, G. D. Matos, J. M. David, G. C. Brandao, E. G. P. da Silva, L. A. Portugal, P. S. dos Reis, A. S. Souza and W. N. L. dos Santos, *Anal. Chim. Acta*, 2007, **597**, 179-186.
29. M. A. Bezerra, R. E. Santelli, E. P. Oliveira, L. S. Villar and L. A. Escaleira, *Talanta*, 2008, **76**, 965-977.

30. S. Ghanbari and G. Kamath, *Energ. Fuels*, 2019, **33**, 5452-5463.
31. V. K. Sangal, V. Kumar and I. M. Mishra, *Comput. Chem. Eng.*, 2012, **40**, 33-40.
32. H. Long, J. Clark, H. Benyounes, W. Shen, L. Dong and S. a. Wei, *Chem. Eng. Technol.*, 2016, **39**, 1077-1086.
33. G. Rionugroho Harvianto, K. Hyun Kim, *Chem. Eng. Res. Des.*, 2019, **144**, 512-519.
34. A. Arjomand, *Iran. J. Chem. Chem. Eng.*, 2018, **37**, 257-270.
35. L. William Luyben, *Chem. Eng. Process.*, 2018, **126**, 132-140.
36. K. J. Kang, G. R. Harvianto and M. Lee, *Ind. Eng. Chem. Res.*, 2017, **56**, 6493-6498.
37. D. Dwivedi, J. P. Strandberg, I. J. Halvorsen, H. A. Preisig and S. Skogestad, *Ind. Eng. Chem. Res.*, 2012, **51**, 15176-15183.
38. N. Van Duc Long and M. Lee, *Comput. Chem. Eng.*, 2012, **37**, 119-124.
39. G. P. Rangaiah, E. L. Ooi and R. Premkumar, *Chemical Product and Process Modeling*, 2009, **1265**, 1934-2659.